# HLA-Cw6 Polymorphism in Autoimmune Blistering Diseases

**DOI:** 10.3390/biom14091150

**Published:** 2024-09-12

**Authors:** Christian Ciolfi, Alvise Sernicola, Mauro Alaibac

**Affiliations:** Dermatology Unit, Department of Medicine (DIMED), University of Padua, 35121 Padova, Italy; chriciolfi96@gmail.com (C.C.); mauro.alaibac@unipd.it (M.A.)

**Keywords:** HLA-Cw6, pemphigus vulgaris, bullous pemphigoid, autoimmune blistering diseases, polymorphism, genotyping

## Abstract

Autoimmune blistering diseases of the pemphigus and pemphigoid groups are immune-mediated disorders due to circulating pathogenetic autoantibodies. Multiple human leukocyte antigen (HLA) genes have been associated with predisposition to these disorders. HLA-Cw6 is involved in antigen presentation processes and has been linked to psoriasis. The aim of our study was to investigate the association between the presence of the HLA-Cw6 allele and susceptibility to pemphigus vulgaris and bullous pemphigoid. A genetic study in vitro with a cross-sectional design was performed enrolling forty patients with pemphigus vulgaris and forty patients with bullous pemphigoid. The detection of HLA-Cw6 was performed through the EUROArray test on DNA obtained from whole blood samples. The polymorphism was detected in 3/40 genotypes in the pemphigus vulgaris group and in 4/40 genotypes of patients with bullous pemphigoid, unveiling a non-statistically significant different frequency in pemphigus (*p* = 0.6368) and in pemphigoid (*p* = 0.62) compared to the reference frequency from the literature of 0.086. Further research is needed to better investigate the role of HLA-Cw6 in immune-mediated diseases and to identify novel genetic markers associated with susceptibility to autoimmune blistering diseases and with disease severity and response to immunosuppressive therapies.

## 1. Introduction

Autoimmune blistering diseases (AIBDs) represent a group of immune-mediated disorders characterized by humoral, as well as cellular, responses resulting in the development of bullae and erosions on the skin and mucous membranes. This is a heterogeneous group of disorders that are clinically challenging as to their diagnosis, requiring the detection of pathogenic autoantibodies in tissue or blood samples, and as to their management, that relies on the use of immunosuppressive agents, which may be an additional source of morbidity, especially in elderly subjects with multiple diseases [1]. Two main groups are recognized: the pemphigus group and the pemphigoid group.

Pemphigus encompasses a group of AIBDs, characterized by intraepidermal acantholysis due to circulating autoantibodies targeting desmogleins 1 (DSG1) and 3 (DSG3), which are transmembrane glycoproteins essential for cell-to-cell adhesion between keratinocytes within desmosomes. Different clinical subtypes are recognized, including pemphigus vulgaris, pemphigus foliaceus, IgA pemphigus, pemphigus herpetiformis, and paraneoplastic pemphigus. Pemphigus vulgaris is the prototypical disease within this class: it is a rare and severe entity that is associated with high mortality of over 20% after 1 year from the diagnosis [2]. Management of pemphigus vulgaris is based on systemic corticosteroids plus the addition of steroid-sparing immunosuppressants; the introduction of the anti-CD20 monoclonal antibody rituximab enhanced the therapeutic armamentarium in this disorder; however, patient selection is a current issue, and potential markers to predict the severe course of disease and treatment response are needed [3,4].

Pemphigoid refers to a group of AIBDs characterized by subepidermal blistering due to circulating autoantibodies targeting two components of the hemidesmosomes that promote epithelial–stromal adhesion: bullous pemphigoid antigen 180 (BP180) and bullous pemphigoid antigen 230 (BP230). Bullous pemphigoid is the most common entity within this group and shows a predilection for elderly individuals: alterations of skin barrier function related to aging contribute to the induction of disease in genetically predisposed individuals. Moreover, immune responses are gradually reshaped with increasing age: immunoregulatory T cell populations characterized by a CD4+ and CD25+ phenotype, which provide a physiological protection against the development of autoimmunity, are depleted while the production of autoantibodies increases. Finally, chronic low-grade inflammation and sustained cytokine production are hallmarks of advanced age and may additionally favor autoimmunity [5,6,7].

It is now acknowledged that genetics plays a major role in the pathogenesis of and individual predisposition to immune-mediated disorders. The human leukocyte antigen (HLA) region, also known as the major histocompatibility complex (MHC), is a highly polymorphous locus in the human genome located on the short arm of chromosome 6 (6p21). Its genes encode proteins involved in immune responses, with class I and class II HLA molecules being the two major classes of HLA antigens. HLA class I molecules are expressed by almost all nucleated cells, while HLA class II molecules are only expressed by antigen-presenting cells; the former present peptides to CD8+ cytotoxic T cells, the latter present peptides to CD4+ T helper cells [8,9].

HLA variants were among the first genetic polymorphisms to be linked to disease susceptibility; in particular, numerous immune-mediated skin conditions have demonstrated evident correlations with distinct HLA class II haplotypes, although the precise mechanisms elucidating how these polymorphisms might impart vulnerability to certain diseases remain largely elusive [10,11,12].

The loss of immunotolerance is considered a key initiating event in the development of AIBDs: specific HLA alleles and single nucleotide polymorphisms, especially those related to Th2 cell activation, are implicated with disease susceptibility, and others are recognized to confer protection [13,14,15].

HLA-Cw6 is one of the most studied psoriasis susceptibility alleles, with psoriasis patients carrying the HLA-Cw6 allele in a proportion ranging from 10.5% to 77.2%. HLA-Cw6 not only confers an increased risk of developing psoriasis but also correlates with a positive family history of psoriasis, an earlier onset of skin lesions, and a more severe course. Furthermore, this allele has also been found to be associated with guttate psoriasis, a higher incidence of the Koebner phenomenon, and a better response to methotrexate and anti-IL12/23 ustekinumab [16].

The aim of our study was to investigate the association between the presence of the HLA-Cw6 allele and susceptibility to pemphigus vulgaris and bullous pemphigoid by comparing the observed frequency of the polymorphism in the two groups of diseases with its prevalence in a healthy reference population derived from the literature [17].

## 2. Materials and Methods

We performed a genetic study in vitro with a cross-sectional design analyzing our hospital electronic medical records database of patients referred to our Dermatology Unit with a diagnosis of AIBD.

Inclusion criteria were the following: diagnosis of pemphigus vulgaris or bullous pemphigoid, confirmed by direct immunofluorescence and serology; availability of an adequate quantity of whole blood sample; age ≥ 18 years; informed consent for study participation provided. To avoid a potential confounding effect of comorbidities, a personal history of other autoimmune disorders and of psoriasis were the exclusion criteria from this study.

Demographic and clinical data, including sex and age at diagnosis, were retrospectively collected for each patient.

Written informed consent for data acquisition was obtained from all eligible patients. Ethical approval was not sought for the present study because it was a collection and analysis of retrospectively obtained and anonymized data for a non-interventional study. Sample size resulted from the number of patients with pemphigus and pemphigoid from which diagnostic material was available.

DNA for genetic analysis was obtained from whole blood samples collected for diagnostic purposes and stored at −80 °C. The detection of HLA-Cw6 was performed through the EUROArray test (Euroimmun Italia, Padua, Italy); this technology relies on the amplification of specific gene sequences through multiplex polymerase chain reaction (PCR), followed by the detection of the resultant PCR products via a hybridization reaction with DNA probes immobilized on a microarray as microscopic spots. As for the HLA-Cw6 test, DNA was isolated from whole blood samples treated with ethylenediaminetetraacetic acid, and a region of the *HLA-C*06* gene was amplified a million-fold through PCR. The resulting PCR products were labeled with a fluorescent dye and then incubated with the microarray containing probes complementary to target DNA. After hybridization, slides were washed and dried prior to analysis of fluorescence using a microarray scanner and EUROArrayScan software version 1.1.12. Test results were positive if any of the 51 detectable subtypes of the HLA-Cw6 gene (*HLA-C*06:02:01:01–C*06:55*) described in the literature was present.

Descriptive statistics were employed to measure the allele frequency of HLA-Cw6 polymorphism, and 95% confidence interval was calculated. Then, we compared the cumulative incidence of the HLA-Cw6 allele reported in a reference population from the literature to the incidence observed in each sample of subjects with either pemphigus vulgaris or bullous pemphigoid. For the hypothesis test on two proportions, significance was set at 5% on a bidirectional z test. Subgroup analysis between HLA-Cw6+ and HLA-Cw6− subjects were performed using the Mann–Whitney U test for quantitative variables and Fisher exact test for qualitative variables. Statistical significance was set for a value of *p* < 0.05. Statistical analyses were performed using Microsoft Excel version 16 (Microsoft Corporation, Redmond, WA, USA).

## 3. Results

Forty patients (15 males and 25 females) with pemphigus vulgaris and 40 patients (19 males and 21 females) with bullous pemphigoid were included in the study.

The HLA-Cw6 polymorphism was detected in 3/40 genotypes in the pemphigus vulgaris group (frequency: 0.075 IC 95% [0.017; 0.133] and in 4/40 genotypes of patients with bullous pemphigoid (frequency: 0.10 IC 95% −0.142; 0.342). The z test showed a non-statistically significant difference in frequency in pemphigus (*p* = 0.6368) and in pemphigoid (*p* = 0.62) compared to the reference frequency from the literature of 0.086. Table 1 summarizes the results of our genetic study.

The demographic and clinical data of patients enrolled in the pemphigus group are listed in Table 2. Sub-group analyses were performed comparing subjects with and without the HLA-Cw6 polymorphism. No statistical difference was observed between the two groups in terms of sex, age at diagnosis, or cutaneous or mucosal involvement.

The demographic and clinical data of patients enrolled in the pemphigoid group are listed in Table 3. Sub-group analyses were performed comparing subjects with and without the HLA-Cw6 polymorphism; similarly to the case of pemphigus, no statistical difference was observed between the two groups in terms of sex, age at diagnosis, and cutaneous or mucosal involvement.

## 4. Discussion and Conclusions

AIBDs of the pemphigus and pemphigoid groups are immune-mediated disorders characterized by circulating pathogenetic autoantibodies. Nonetheless, there is an increasing acknowledgment of the pivotal contribution of cell-mediated immunity in the pathogenesis of such diseases. In particular, T cells promote the survival and differentiation of self-reactive B cells, thereby augmenting antibody production; moreover, they contribute to inflammation and tissue damage through the secretion of pro-inflammatory cytokines [18]. Specifically, Th1 cells enhance immune responses through interferon-gamma; Th2 cells promote B cell proliferation and production of pathogenic autoantibodies. Th17 cells secrete IL-17, contributing to inflammation and tissue damage, while regulatory T cells inhibit desmoglein-3-autoreactive T cells and antibody production. Finally, T follicular helper cells facilitate autoantibody production through a cross-talk with B cells [18].

These potential shared mechanisms between psoriasis and AIBDs provide the rationale for investigating a possible correlation with the HLA-Cw6 allele in AIBDs.

HLA-Cw6 is one of the most strongly associated psoriasis susceptibility alleles. This allele can trigger a cell-mediated immune response via antigen presentation to T cells. Two autoantigens have been identified so far, LL-37 and ADAMTSL5, both overexpressed in psoriatic lesions: LL-37 is a cathelicidin-derived antimicrobial peptide which forms a complex with self- or non-self-DNA/RNA and is presented via HLA-Cw6 to T cells and plasmacytoid dendritic cells; ADAMTSL5 (a disintegrin-like and metalloprotease domain containing thrombospondin type 1 motif-like 5) is a melanocyte-derived protein capable of activating Th17 cells. Not only is HLA-Cw6 associated with an increased risk of developing psoriasis, but it also correlates with a positive family history for psoriasis and a more severe and unstable disease. Finally, this allele has been found to predict a better response to methotrexate and anti-IL12/23 ustekinumab [16,19].

Notably, multiple HLA genes have been associated with predisposition to AIBDs development; recently, a review of the role of HLA class II in AIBDs has been performed by the authors of the present study [20]. Regarding bullous pemphigoid, the *HLA-DQB1*0301* is the most strongly associated allele in Caucasians and other ethnic groups, including Brazilians and Iranians. A recent meta-analysis highlighted that *HLA-DQA1*0505* is associated with an increased risk of bullous pemphigoid, while *DQA1*0201* is protective against the disease [21]. In pemphigus, several HLA types exhibit population specificity, while others are linked to pemphigus across multiple ethnicities. Among the latter are *HLA-DQB1*0503* and *HLA-DRB1*0402*. Other very common alleles belong to the HLA-A10, B38, DR4, and DQw3.2 haplotypes; individuals lacking the DR4 haplotype usually carry HLA-DRw6 [22]. Several studies have also identified associations between pemphigus and HLA alleles in specific ethnic populations. Certain HLA haplotypes render patients genetically susceptible to pemphigus and predict a more severe course of the disease. For instance, Vietnamese patients carrying the *HLA-DRB1*04* alleles were more likely to have mild or moderate pemphigus [23], whereas a correlation of the *HLA-DRB1*04:02* and *DQB1*03:02* alleles with more severe pemphigus was highlighted in a group of 44 patients from Slovakia [24]. In addition to HLA haplotypes, single-nucleotide polymorphisms (SNPs) have recently emerged as new potential markers that may contribute not only to the development of AIBDs but also to their phenotypic variability and therapeutic outcomes. SNPs are the most common types of genetic variation and are characterized by a genomic DNA variant at a single base position. The role of SNPs in pemphigus has been extensively reviewed by Mahmoudi et al. in 2020 [25]. To the best of our knowledge, this is the first study to investigate the association between HLA-Cw6 and AIBDs.

In our cohort of patients, we found an HLA-Cw6 genotype frequency of 7.5% in the pemphigus group and 10% in the pemphigoid group, but no statistically significant difference was found compared to the prevalence reported in the literature for an Italian population. No patient from the HLA-Cw6+ groups in our study has been treated with methotrexate; therefore, we are unable to provide information on a potential predictive role of this allele towards methotrexate response in pemphigus and pemphigoid, as occurs in the context of psoriasis. Limitations of the present study to be considered are the small sample size, due to the rarity of AIBDs, and the retrospective design with consequent lack of data regarding treatment response and disease progression over time. Future larger studies may also highlight additional associations with severity of disease or specific clinical phenotypes. The EUROarray targeted genotyping technique was chosen for the detection of HLA-Cw6 alleles from whole blood for this study considering its suitability for the specific detection of known polymorphisms in a clinical laboratory setting with 100% sensitivity and specificity, according to the manufacturer, in respect to the reference molecular genetic method. However, the EUROarray technique has certain disadvantages compared to next-generation sequencing technologies. The latter include HLA deep sequencing approaches that can detect novel polymorphisms and complex variants, which would remain undetectable by a targeted array; presently, the accessibility to deep sequencing technologies is limited by the cost of the test and complexity of the generated data. Finally, the targeted and specific array that was employed in this study allowed the detection of the HLA-Cw6 polymorphisms but not of additional HLA polymorphisms that have been related to AIBDs in the literature. Future studies may include these genes to provide useful positive controls in subjects with AIBDs.

Further larger studies are certainly needed to better define the role of HLA-Cw6 in immune-mediated diseases like AIBDs and to find new potential genetic markers of susceptibility to AIBDs, as well as predictors of disease severity and response to immunosuppressive therapies.

## Figures and Tables

**Table 1 biomolecules-14-01150-t001:** Genotype count of HLA-Cw6+ polymorphism in subjects with pemphigus vulgaris and bullous pemphigoid.

Group	HLA-Cw6+	*p*-Value *
PV	0.075 [0.017; 0.133]	0.6368
PB	0.10 [−0.142; 0.342]	0.62
Healthy population	0.086	

* The reported *p*-values are calculated with bidirectional z test. Abbreviations: PB, bullous pemphigoid; PV, pemphigus vulgaris.

**Table 2 biomolecules-14-01150-t002:** Demographic and clinical characteristics of patients with pemphigus vulgaris according to HLA-Cw6+ and HLA-Cw6− subgroups.

Variable	HLA-Cw6+	HLA-Cw6−	*p*-Value *
Number of patients	3 (7.5%)	37 (92.5%)	
Sex	F = 3 (100%)	22 (59.46%)	
	M = 0	15 (40.54%)	0.2788
Age at diagnosis [median (min; max)]	53 (40; 66) years	50 (20; 89) years	0.9045
Type of involvement	Skin = 2 (66.67%) Mucosae = 3 (100%)	25 (67.57%) 28 (75.68%)	1 0.5655

* The reported *p*-values are calculated with Mann–Whitney U test for age and Fisher exact test for sex and type of involvement.

**Table 3 biomolecules-14-01150-t003:** Demographic and clinical characteristics of patients with bullous pemphigoid according to HLA-Cw6+ and HLA-Cw6− subgroups.

Variable	HLA-Cw6+	HLA-Cw6−	*p*-Value *
Number of patients	4 (10%)	36 (90%)	
Sex	F = 2 (50%)	18 (50%)	
	M = 2 (50%)	18 (50%)	1
Age at diagnosis [median (min; max)]	62.5 (57; 95) years	66.5 (41; 91) years	0.8026
Type of involvement	Skin = 4 (100%) Mucosae = 2 (50%)	Skin = 36 (100%) Mucosae = 4 (11.11%)	1 0.0997

* The reported *p*-values are calculated with Mann–Whitney U test for age and Fisher exact test for sex and type of involvement.

## Data Availability

The data that support the findings of this study are available from the corresponding author upon reasonable request.

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
