# Peer review of "HLA-Cw6 Polymorphism in Autoimmune Blistering Diseases"

_biomolecules, 2024, doi:10.3390/biom14091150_

Round 1
Reviewer 1 Report
Comments and Suggestions for Authors
The current manuscript presents data regarding the association of HLA-Cw6 with autoimmune bullous diseases (AIBDs), specifically pemphigus vulgaris (PV) and bullous pemphigoid (BP). Although HLA-Cw6 has been strongly linked to psoriasis, this study did not find a significant difference in HLA-Cw6 frequencies in these AIBDs compared to those in normal controls. However, there are some areas that need to be addressed in this study:
1) It is important for the study to provide clear exclusion and inclusion criteria. Specifically, does the patient group have a medical history of other autoimmune diseases? If comorbid psoriasis is present, a detailed description is necessary.
2) The method used to detect HLA-Cw6, EUROArray, involves PCR of the genomic DNA from whole blood followed by binding to probes complementary to target sequences. However, the discussion section should include a brief description of the sensitivity and specificity for detecting HLA-Cw6. Additionally, it would be beneficial to discuss the advantages and disadvantages of this approach compared to HLA deep sequencing.
3) Some HLA genes, such as DRB1*0402 and DQB1*0503, are closely correlated with AIBDS. Considering the negative findings for HLA-Cw6, these genes should serve as better positive controls. It's suggested that the author provide these data.
4) Line 128 should specify the HLA-Cw6 polymorphism and detail the genotypes for HLA-Cw6 of the patients.
5) In Table 1, the HLA-Cw6 frequency in the healthy population is listed as 0.078, which is inconsistent with the number (0.086) in Line 132.
6) The statistical methods used for testing the significance between the healthy population and the PV or BP group should be clearly stated as a one-sample t-test or z-test (Table 1). The specific method used for testing should be provided in the figure or table legends.
Comments on the Quality of English LanguageThe English language appears OK, requiring only minor editing.
Author Response
Reviewer 1
Comments and Suggestions for Authors
The current manuscript presents data regarding the association of HLA-Cw6 with autoimmune bullous diseases (AIBDs), specifically pemphigus vulgaris (PV) and bullous pemphigoid (BP). Although HLA-Cw6 has been strongly linked to psoriasis, this study did not find a significant difference in HLA-Cw6 frequencies in these AIBDs compared to those in normal controls. However, there are some areas that need to be addressed in this study:
Dear Reviewer,
Thank you for your careful review of our manuscript and for your constructive comments. We have carefully considered each issue raised and our responses are provided below point by point.
1) It is important for the study to provide clear exclusion and inclusion criteria. Specifically, does the patient group have a medical history of other autoimmune diseases? If comorbid psoriasis is present, a detailed description is necessary.
Thank you, we have clarified this aspect in the methods. Please see in the text: “To avoid a potential confounding effect of comorbidities, a personal history of other au-toimmune disorders and of psoriasis were the exclusion criteria from this study.”
2) The method used to detect HLA-Cw6, EUROArray, involves PCR of the genomic DNA from whole blood followed by binding to probes complementary to target sequences. However, the discussion section should include a brief description of the sensitivity and specificity for detecting HLA-Cw6. Additionally, it would be beneficial to discuss the advantages and disadvantages of this approach compared to HLA deep sequencing.
Thank you for your comment, a relevant statement was added to the text. Please see in the discussion: “The EUROarray targeted genotyping technique was chosen for the detection of HLA-Cw6 alleles from whole blood for this study considering its suitability for the specific detection of know polymorphisms in a clinical laboratory setting with 100% sensitivity and specificity, according to the manufacturer, in respect to a reference molecular genetic method. However, the EUROarray technique has certain disadvantages compared to next generation sequencing technologies. The latter include HLA deep sequencing approaches that can detect novel polymorphisms and complex variants, which would remain undetectable by a targeted array; presently, the accessibility to deep sequencing technologies is limited by the cost of the test and complexity of the generated data.”
3) Some HLA genes, such as DRB1*0402 and DQB1*0503, are closely correlated with AIBDS. Considering the negative findings for HLA-Cw6, these genes should serve as better positive controls. It's suggested that the author provide these data.
Thank you, we have added an explanatory sentence to the discussion. Please see in the text: “Finally, the targeted and specific array that was employed in this study allowed the detection of the HLA-Cw6 polymorphisms but not of additional HLA polymorphisms that have been related to AIBDs in the literature. Future studies may include these genes to provide useful positive controls in subjects with AIBDs.”
4) Line 128 should specify the HLA-Cw6 polymorphism and detail the genotypes for HLA-Cw6 of the patients.
Thank you for your comment, we have added a sentence to explain the test results in the methods section. Please see in the text: “Test results were positive if any of the 51 detectable subtypes of the HLA-Cw6 gene (HLA-C*06:02:01:01 – C*06:55) described in the literature was present.”
5) In Table 1, the HLA-Cw6 frequency in the healthy population is listed as 0.078, which is inconsistent with the number (0.086) in Line 132.
We apologize for this mistake in the table; the value is now corrected to 0.086.
6) The statistical methods used for testing the significance between the healthy population and the PV or BP group should be clearly stated as a one-sample t-test or z-test (Table 1). The specific method used for testing should be provided in the figure or table legends.
Thank you for your suggestion. We have added the following explanations below table 1 (“*The reported p-values are calculated with bidirectional z test.”) and table 2 and 3 (“*The reported p-values are calculated with Mann-Whitney U test for age and Fisher exact test for sex and type of involvement.”)
Reviewer 2 Report
Comments and Suggestions for Authors
Dear Authors,
I agree with the proposed format, length, and structure of the manuscript.
Please take into consideration few suggestions:
Line 9 – replace “gens” with “genes”
Line 42 – despite the availability of Rtx, corticosteroids are still widely and mainly used in the treatment of pemphigus and I wouldn’t determine their use as “historic”, so please remove or replace that term
Line 43 – please write “systemic corticosteroids”
Line 165 – avoid phrase repetition: “of the most” is written twice
References
The review of the literature on the topic is exhaustive enough.
I would suggest removing ref 15:
15. Namazi, M.R. Oral Sex May Afford Protection against Pemphigoid Gestationis. Med Hypotheses 2007, 69, 1386–1387, 238 doi:10.1016/j.mehy.2006.12.038
Instead, I would suggest citing another article dedicated to genetics of pemphigus, namely:
Drenovska K, Ivanova M, Vassileva S, Shahid MA, Naumova E. Association of specific HLA alleles and haplotypes with pemphigus vulgaris in the Bulgarian population. Front Immunol. 2022 Aug 2;13:901386. doi: 10.3389/fimmu.2022.901386. PMID: 35983062; PMCID: PMC9378788.
Author Response
Reviewer 2
Dear Authors,
I agree with the proposed format, length, and structure of the manuscript.
Dear Reviewer,
thank you for your efforts on our manuscript and for your comments. Please find our responses provided below point by point.
Please take into consideration few suggestions:
Line 9 – replace “gens” with “genes”
Thank you. We have changed “gens” into “genes”.
Line 42 – despite the availability of Rtx, corticosteroids are still widely and mainly used in the treatment of pemphigus and I wouldn’t determine their use as “historic”, so please remove or replace that term
Thank you for your consideration. We agree that, beyond rituximab, corticosteroids are still the mainstay of management both for induction and maintenance therapy of pemphigus vulgaris. We have removed the term “historic”.
Line 43 – please write “systemic corticosteroids”
Thank you. We have modified “corticosteroid” into “corticosteroids”.
Line 165 – avoid phrase repetition: “of the most” is written twice
Thank you. We have removed the repeated phrase.
References
The review of the literature on the topic is exhaustive enough.
I would suggest removing ref 15:
- Namazi, M.R. Oral Sex May Afford Protection against Pemphigoid Gestationis. Med Hypotheses 2007, 69, 1386–1387, 238 doi:10.1016/j.mehy.2006.12.038
Instead, I would suggest citing another article dedicated to genetics of pemphigus, namely:
Drenovska K, Ivanova M, Vassileva S, Shahid MA, Naumova E. Association of specific HLA alleles and haplotypes with pemphigus vulgaris in the Bulgarian population. Front Immunol. 2022 Aug 2;13:901386. doi: 10.3389/fimmu.2022.901386. PMID: 35983062; PMCID: PMC9378788.
Thank you for your suggestion. We have changed reference 15 to “Drenovska, K.; Ivanova, M.; Vassileva, S.; Shahid, M.A.; Naumova, E. Association of Specific HLA Alleles and Haplotypes with Pemphigus Vulgaris in the Bulgarian Population. Front Immunol 2022, 13, 901386, doi:10.3389/fimmu.2022.901386.”
Reviewer 3 Report
Comments and Suggestions for Authors
First, I would like to congratulate the authors on their work. Despite their rarity, autoimmune blistering disorders are associated with significant morbidity and high mortality rates even with novel treatments. The identification of biomarkers predictive of disease severity and response to various treatment strategies is an urgent need in this patient population. Therefore, studies like the one carried out by the authors are extremely valuable.
The structure of the article is classic, scientifically correct. The presentation is concise and the style clear and accessible. The references are relevant and recent. The novelty of the research is unquestionable, this being the first study to investigate the association between HLA Cw6 and autoimmune blistering diseases.
However, I have a few suggestions.
I recommend expanding the discussion regarding the genetic background of autoimmune blistering diseases, focusing on both specific and nonspecific genetic factors.
Several studies have investigated the role of genetic variations, especially single nucleotide polymorphisms in the pathogenesis of autoimmune blistering diseases, as well as their associations with disease severity and response to treatment. A synthetic presentation of these studies would place the current research in a wider context.
The authors mentioned the lack of data regarding treatment response and disease progression over time due to the retrospective nature of the study. Nonetheless, it would be interesting if the authors could provide information related to the severity of the disease or the particularities of the disease in HLA Cw6+ patients, as such data are readily available from the patients’ records.
The authors refer to the increased susceptibility of HLA-Cw6+ psoriasis patients to methotrexate treatment. Methotrexate is also used in the treatment of patients diagnosed with pemphigus and other autoimmune blistering disorders. Did HLA-Cw6+ patients identified by the authors receive such a treatment? I encourage the authors to address this issue and comment on the potential value of HLA-Cw6 as a predictor of a better response to methotrexate in pemphigus patients.
Best regards!
Comments on the Quality of English LanguageMinor editing is required.
Author Response
Reviewer 3
First, I would like to congratulate the authors on their work. Despite their rarity, autoimmune blistering disorders are associated with significant morbidity and high mortality rates even with novel treatments. The identification of biomarkers predictive of disease severity and response to various treatment strategies is an urgent need in this patient population. Therefore, studies like the one carried out by the authors are extremely valuable.
The structure of the article is classic, scientifically correct. The presentation is concise and the style clear and accessible. The references are relevant and recent. The novelty of the research is unquestionable, this being the first study to investigate the association between HLA Cw6 and autoimmune blistering diseases.
Dear Reviewer,
Thank you for your careful revision of our manuscript and for your encouraging comments. Please find our responses below.
However, I have a few suggestions.
I recommend expanding the discussion regarding the genetic background of autoimmune blistering diseases, focusing on both specific and nonspecific genetic factors.
Several studies have investigated the role of genetic variations, especially single nucleotide polymorphisms in the pathogenesis of autoimmune blistering diseases, as well as their associations with disease severity and response to treatment. A synthetic presentation of these studies would place the current research in a wider context.
Thank you for your constructive comment. The following paragraph was added in the discussion section: “Regarding bullous pemphigoid, the HLA-DQB1*0301 is the most strongly associated allele in Caucasians and other ethnic groups, including Brazilians and Iranians. A recent meta-analysis highlighted that HLA-DQA1*0505 is associated with an increased risk of bullous pemphigoid, while DQA1*0201 is protective against the disease [21]. In pem-phigus, several HLA types exhibit population specificity, while others are linked to pemphigus across multiple ethnicities. Among the latter are HLA-DQB1*0503 and HLA-DRB1*0402. Other very common alleles belong to the HLA-A10, B38, DR4, DQw3.2 haplotype; individuals lacking the DR4 haplotype usually carry HLA-DRw6 [22]. Several studies have also identified associations between pemphigus and HLA alleles in specific ethnic populations. Certain HLA haplotypes render patients genetically susceptible to pemphigus and predict a more severe course of the disease. For instance, Vietnamese patients carrying the HLA-DRB1*04 alleles were more likely to have mild or moderate pemphigus [23], whereas a correlation of the HLA DRB1*04:02 and DQB1*03:02 alleles with more severe pemphigus was highlighted in a group of 44 patients from Slovakia [24]. In addition to HLA haplotypes, single-nucleotide polymorphisms (SNPs) have recently emerged as new potential markers which may contribute not only to the development of AIBDs, but also to their phenotypic variability and therapeutic outcomes. SNPs are the most common types of genetic variation and are characterized by a genomic DNA variant at a single base position. The role of SNPs in pemphigus has been extensively reviewed by Mahmoudi et al. in 2020 [25].
The authors mentioned the lack of data regarding treatment response and disease progression over time due to the retrospective nature of the study. Nonetheless, it would be interesting if the authors could provide information related to the severity of the disease or the particularities of the disease in HLA Cw6+ patients, as such data are readily available from the patients’ records.
Thank you for your suggestion. We chose not to report additional clinical variables apart from the age at diagnosis and type of involvevement as we were not able to highlight any differences.
We added the following statement to the limitations of our study in the discussion section. Please see: “Future larger studies may also highlight additional associations with severity of disease or specific clinical phenotypes.”
The authors refer to the increased susceptibility of HLA-Cw6+ psoriasis patients to methotrexate treatment. Methotrexate is also used in the treatment of patients diagnosed with pemphigus and other autoimmune blistering disorders. Did HLA-Cw6+ patients identified by the authors receive such a treatment? I encourage the authors to address this issue and comment on the potential value of HLA-Cw6 as a predictor of a better response to methotrexate in pemphigus patients.
Best regards!
Thank you for your consideration. None of the Cw6+ patients were treated with methotrexate, but we agree that HLA-Cw6 may predict a better response to methotrexate in pemphigus vulgaris patients as well, and this aspect should be investigated in future studies.
A sentence about this topic has been added to the Discussion accordingly. Please see in the text: “No patient from the HLA-Cw6+ groups in our study has been treated with methotrexate; therefore, we are unable to provide information on a potential predictive role of this allele towards methotrexate response in pemphigus and pemphigoid, as occurs in the context of psoriasis.”
Round 2
Reviewer 1 Report
Comments and Suggestions for Authors
The report presented negative findings for the HLA-Cw6 polymorphism in pemphigus and BP compared to the controls. However, the results also seem informative and novel. The revised manuscript has addressed all the concerns in my review, making it now acceptable.
Comments on the Quality of English LanguageThe English language is acceptable in this revised manuscript.